# The Role of the Human Cytomegalovirus *UL133-UL138* Gene Locus in Latency and Reactivation

**DOI:** 10.3390/v12070714

**Published:** 2020-07-01

**Authors:** Luwanika Mlera, Melissa Moy, Kristen Maness, Linh N. Tran, Felicia D. Goodrum

**Affiliations:** 1BIO5 Institute, University of Arizona, Tucson, AZ 85719, USA; luwanikamlera@arizona.edu; 2Graduate Interdisciplinary Program in Cancer Biology, Tucson, AZ 85719, USA; mmoy@email.arizona.edu; 3Immunobiology Department, University of Arizona, Tucson, AZ 85719, USA; kmaness@email.arizona.edu (K.M.); lnt@email.arizona.edu (L.N.T.)

**Keywords:** Human cytomegalovirus, herpesvirus, DNA virus, latency, reactivation, *UL135*, *UL136*, *UL138*

## Abstract

Human cytomegalovirus (HCMV) latency, the means by which the virus persists indefinitely in an infected individual, is a major frontier of current research efforts in the field. Towards developing a comprehensive understanding of HCMV latency and its reactivation from latency, viral determinants of latency and reactivation and their host interactions that govern the latent state and reactivation from latency have been identified. The polycistronic *UL133-UL138* locus encodes determinants of both latency and reactivation. In this review, we survey the model systems used to investigate latency and new findings from these systems. Particular focus is given to the roles of the UL133, UL135, UL136 and UL138 proteins in regulating viral latency and how their known host interactions contribute to regulating host signaling pathways towards the establishment of or exit from latency. Understanding the mechanisms underlying viral latency and reactivation is important in developing strategies to block reactivation and prevent CMV disease in immunocompromised individuals, such as transplant patients.

## 1. Introduction

Human cytomegalovirus (HCMV) is a ubiquitous beta-herpesvirus infecting up to 99% of the world’s population, depending on geographic location [1,2]. Transmission is mediated by direct contact with urine, saliva or other bodily fluids from individuals that are actively shedding the virus [3]. Infection of an immunocompetent host does not typically lead to overt clinical disease. However, the virus is never cleared; instead it persists in a latent state whereby viral genomes are maintained without infectious virus production. While HCMV infects and replicates in many different cell types, latency is established in hematopoietic progenitor cells and myeloid lineage cells [4,5,6]. Changes in the cellular environment (e.g., stress or differentiation) or changes in physiology of the host may result in reactivation of the virus. HCMV reactivation is thought to happen frequently and sporadically, albeit subclinically, in immunocompetent individuals.

HCMV reactivation and/or reinfection with a new strain is a major risk factor for severe morbidity and mortality in immunocompromised patients. HCMV reactivation continues to be a significant cause of death in solid organ and stem cell transplant recipients and can cause life-threatening disease in cancer patients undergoing intensive chemotherapy regimens that weaken immune responses. HCMV reactivation increases the time and cost of hospital stays for transplant patients [7,8,9]. The severity of disease in stem cell transplant patients is exacerbated because HCMV is myelosuppressive [10], delaying engraftment and increasing susceptibility to infection due to a reduction in leukocytes, as well as red blood cells and platelets. HCMV also has onco-modulatory properties and viral activation of cellular pathways may increase cell proliferation, tumor growth, enhanced cell survival and angiogenesis [11,12].

HCMV is also the leading infectious agent responsible for congenital birth defects [13,14,15]. 1 in 150 children born in the US acquires the HCMV in utero. Outcomes of congenital HCMV infection include hearing loss, other mild to severe neurological and cognitive deficits (e.g., developmental delay in motor skills), and microcephaly. While many infected babies are asymptomatic at birth, approximately 20% of the infected babies will have permanent disability. While effects of the congenital infection may be evident at birth, some are not realized until later in infancy or childhood. The most severely affected children suffer recurrent seizures. Considering the congenital infection and infection in the immune compromised, the public health impact of HCMV is immense.

Current antiviral therapies include inhibitors of the viral DNA polymerase (e.g., ganciclovir, foscarnet, cidofovir), the viral kinase (e.g., maribivir) or the terminase (e.g., letermovir) offer limited effectiveness due to the toxicity of some, and only target cells actively replicating virus. New rationally designed therapeutic agents are required, and those that can target latent infection would offer great advantage in combatting CMV reactivation and disease. Achieving this will require understanding of the complex mechanisms associated with the biology of HCMV latency and reactivation.

Over the last decade, focus has been turned to defining viral determinants of latency and the host pathways they target to delineate the molecular mechanisms underlying HCMV latency and reactivation. A major focus of the research has been on the role of a locus of genes in the UL*b*’ region coined the *UL133-138* locus. This polycistronic locus contains genes that influence whether the virus will replicate or establish latency and also play key roles in the “decision” to reactivate to generate new infectious virus. Further, the field is beginning to understand how the virus has hardwired this locus into host biology to allow the virus to “sense” and “respond” to host cues. This review will consider the role of each of the *UL133-138* genes and their role collectively in regulating states of HCMV latency and reactivation. Knowledge gaps and future prospects are discussed with the aim of stimulating new interest and additional research into this locus.

## 2. Experimental Models for HCMV Latency and Reactivation

A comprehensive understanding of HCMV latency has been a slow study due to the restriction of HCMV to the human species and the restriction of HCMV latency to primary hematopoietic progenitor cells and some cells lines. A very small number of cells in the human host harbor latent HCMV infection [16]. Further, reactivation is spontaneous and unpredictable because the biochemical and biophysical “stress” factors that trigger virus reactivation are poorly defined. Therefore, in vitro cell culture models have been fundamental to investigating the biology of latent HCMV infection, despite their inherent limitations (e.g., heterogeneity, limited starting material, spontaneous differentiation).

Primary CD34+ hematopoietic progenitor cells (HPCs) derived from human bone marrow or cord blood are an invaluable tool used to recapitulate latent HCMV infection in vitro [17]. For latency, the sensitivity to culture conditions and the proclivity of CD34+ HPCs to spontaneously differentiate in culture must be carefully considered. Stromal cell clones are commonly used to maintain progenitor populations with in vivo engraftment and hematopoietic reconstitution potential and to support physiologically relevant hematopoietic differentiation [17,18,19,20,21,22]. However, other models are also common that culture progenitor populations in cell-free systems with the addition of cytokines [23,24,25]. Once infected, CD34+ HPCs maintain HCMV genomes in the absence of productive replication. However, the heterogeneity of this population can result in the loss of genomes from some cells and other cells that spontaneously replicate virus. Therefore, it is important to monitor the maintenance of genomes and production of virus progeny prior to applying a reactivation stimulus so that the baseline for replication during the “latent” period is known. HCMV reactivation, which is estimated to occur in about 1 in 9000 cells infected in vitro, is induced by co-culturing infected CD34+ HPCs with fibroblasts in cytokines to promote myeloid differentiation of the hematopoietic cells [17]. Apart from the relatively high cost and limiting material associated with CD34+ HPCs, the model is robust, and it has been widely adopted as the gold standard for understanding latency. It has been powerful in quantitatively defining phenotypes of recombinant viruses for their ability to establish latency or reactivate from latency [26]. While other latent reservoirs may exist, but are yet unidentified in the human host, latency studies have remained in hematopoietic cells as most other cell types infected support at least low levels of productive replication.

Latent HCMV genomes are carried as HPCs differentiate down the monocytic lineage. Reactivation is known to occur upon differentiation of these cells into macrophages or dendritic cells [5,24,27,28,29,30] and monocytes play an important role in dissemination of the virus by carrying virus into tissues [31,32]. Monocytes are readily infected in culture and while infection of monocytes has been described as a quiescent state as opposed to a latent state, direct infection of monocytes is also used as a model of latency [23,33,34,35,36]. It is currently unknown how infection differs in cells infected directly as monocytes, possibly through contact with infected tissue as they circulate, compared to monocytes that were infected as progenitors and carried the viral genome as they differentiated to monocytes. HCMV infection both directly and indirectly limits hematopoietic differentiation potential [37,38], and it has recently been shown that infected CD34+ HPCs secrete transforming growth factor beta (TGFβ), which impedes the differentiation of bystander hematopoietic cells, providing a mechanistic insight into myelosuppression [39]. Intriguingly, viral miRNAs prevent TGFβ signaling from impacting the infected cell. Further, UL7 is expressed during reactivation and stimulates differentiation of both CD34+ HPCs and CD14+ monocytes [40].

The THP-1 (developed at Tohoku Hospital of Pediatrics, Japan) cell line is a spontaneously immortalized monocytic leukemia cell line [41]. The cell line is used as a model for human monocytes [42], and also utilized as an HCMV latency model [43,44,45,46]. Upon infection, THP-1 cells undergo a latent-like HCMV infection and reactivation is activated by treatment with a phorbol ester, 12-*O*-tetradecanoylphorbol-13-acetate or TPA [44]. TPA treatment differentiates THP-1 cells from non-adherent monocytic cells to an adherent macrophage-like phenotype. Reactivation is measurable by increases in immediate early 1 and 2 (IE1-72 kDa and IE2-86 kDa), early and late gene products, although genome amplification and virus production are modest at best. These cells offer the advantage that reactivation of gene expression is synchronous and occurs at a frequency greater than that of primary cells in culture and have been important for studying silencing and transcriptional re-expression of viral genes [43]. Other cell line models for latency exist, including Kasumi-3 and Ntera2 cells that similarly recapitulate aspects of latency [47,48].

Humanized mouse models (huNSG), which support latent HCMV in vivo, overcome the lack of an animal model for HCMV latency and reactivation [40,49,50,51]. HuNSG mice are sub-lethally irradiated and engrafted with CD34+ HPCs. Infection is established by intraperitoneal injection with HCMV-infected fibroblast at 5 weeks post-engraftment [49,50]. Following an additional 5 weeks, reactivation is stimulated by granulocyte colony stimulating factor (G-CSF) delivered using a subcutaneous pump [49,50]. G-CSF treatment mobilizes HCMV-infected CD34+ HPCs into the peripheral blood and tissues causing reactivation [49,52]. HCMV genome levels in blood, bone marrow and organs are used as an indicator of reactivation 7 days following the G-CSF stimulus [49,50]. The huNSG model has been used to recapitulate in vitro phenotypes observed with HCMV recombinants that have mutations in the *UL133-138* gene locus [51,53].

## 3. Historical Perspective and Integration of Model Systems

At 236 kb, HCMV has the largest genome among the herpesviruses, and it encodes at least 200 open reading frames [54]. The UL*b*’ region of the genome encodes ~20 viral proteins present in the genomes of clinical HCMV isolates and lost during serial passage of the virus in cultured fibroblasts [55]. The absence of the UL*b*’ region in passaged HCMV strains (laboratory-adapted) indicates that the UL*b*’ region is not required for replication.

Interest in ULb’ genes with regards to latency stemmed from the unknown, but likely roles they might play in other contexts of infection, and particularly latency. Indeed, viruses lacking the ULb’ region replicate to higher titers compared to clinical isolates [56], suggesting that the UL*b*’ genes confer a restriction on virus replication, in addition to roles in modulating latency and immune evasion. Functions for UL*b*’ genes remained largely undefined as there had been no previous characterization of the vast majority of UL*b*’ genes before 2000. Large 5-kb and single ORF deletions across the UL*b*’ region in a bacterial artificial chromosome clone of HCMV (FIX strain) identified sequences associated with altered ability to replicate relative to the WT virus in primary human CD34+ HPCs [17,57]. Of note, viruses containing deletions encompassing UL151-UL135 were severely defective for reconstitution of the virus from transfection of infectious genomes into fibroblasts. By contrast, a virus containing another 5-kb deletion encompassing UL136-UL142 had a replication advantage in fibroblasts, and was more replicative in CD34+ HPCs even in the absence of a reactivation stimulus, indicating that this virus failed to establish latency [57]. From the use of recombinant viruses containing substitutions or stop codon insertions to disrupt single ORFs, it is now known that the growth restriction of the ∆UL150-UL135 mutant virus is due to the loss of UL135 and the replicative advantage of ∆UL136-UL142 is due to the loss of UL138. The requirement for UL135 was surprising as the entire UL*b*’ region could be deleted without a cost to replication in fibroblasts [55]. As discussed in detail below, progress was made to understand that UL135 and UL138 have an antagonistic relationship [19]. UL135 is required to overcome a virus-coded suppression of replication that is conferred by UL138, and the defect in replication of the ∆UL135 virus could largely be overcome by subsequent disruption of *UL138*.

As important as the goal of understanding latency, is the goal of simply understanding the basic biology of these viral genes and their gene products. Towards this prevailing goal, a variety of model systems have been used to define the biology of gene products. The starting material required for many molecular and biochemical approaches, and the relative difficulty of manipulating primary human hematopoietic cells in culture (e.g., overexpression or knockdown) are formidable challenges to studies pertaining to latency and force the requirement for additional model systems. Fibroblasts have been the workhorse of the field for investigating the productive infection. Fibroblasts have been useful in defining the functional roles of and interactions between *UL133-UL138* genes in infection and to begin identifying host interactions. Endothelial cells, which support a low-level, smoldering or chronic infection, have been a particularly informative model for understanding the requirements for UL135 and UL136 for replication in this context [51,58]. The huNSG model has been important to confirm in vitro phenotypes and, in the case of UL136, reveal context-dependent differences [51,53]. A combined approach using each of these model systems—fibroblasts, endothelial cells, CD34+ HPCs and huNSG animals—has provided robust systems for defining the biology and function associated with viral genes at the focus of this review.

Strikingly, this combined approach has demonstrated conservation in the function of these proteins across contexts, although the significance of these functions for the outcome of infection might be very different in each context of infection. As an example, UL135 downregulates EGFR and viruses lacking UL135 or containing point mutations in UL135 that disrupt its interaction with host factors Abi-1 and CIN85 have increased levels of surface and total levels of EGFR in both fibroblasts and CD34+ HPCs [59,60]. While the UL135/Abi-1/CIN85 interactions affect EGFR surface levels in both fibroblasts and CD34+ HPCs, abolishing these interactions affects replication (e.g., reactivation from latency) in CD34+ HPCs or in HuNSG, but not in fibroblasts [53,59,60]. Further, inhibition of EGFR signaling, or downstream pathways moderately enhances replication in fibroblasts (~5-fold), and it robustly rescues reactivation of UL135-null viruses in CD34+ HPCs. Therefore, while the UL135/Abi-1/CIN85/EGFR interaction is best defined in fibroblasts, the function is conserved and importance amplified in CD34+ HPCs.

Why then do CD34+ HPCs support latency whereas fibroblast support only replicative infection if UL133-UL138 proteins function similarly in both cells? Afterall, it might be tempting to speculate that cellular pathways would be regulated in opposite directions in cells that support latency versus those that support reactivation. While the simplicity of this possibility is attractive, it is harder to imagine how a viral protein regulating these pathways might exert opposite effects on the same target. An alternative possibility is that viral proteins function very similarly in most contexts, but it is the cell biology that dictates different outcomes. The functional and phenotypic differences observed for ULb’ genes between fibroblasts and CD34+ HPC or any other cell types may be due to the differences in the cell biology inherent to each context of infection and the resulting profile of viral gene expression. Both fibroblasts and hematopoietic cells express a broad proportion of the viral genome during infection, but transcripts accumulate to much greater levels in fibroblasts as compared to CD34+ HPCs [61,62]. UL*b*’ genes that are largely dispensable for replication in fibroblasts have more pronounced effects on infection in CD34+ HPCs where gene expression is much lower than in fibroblasts [62]. It is likely then that latency might be able to be established and maintained in hematopoietic cells, whereas high levels of IE, early and late gene expression in fibroblasts overcomes the suppressive affects of genes such as UL138.

It is also possible that antagonistic genes such as UL135 and UL138 are differentially regulated in different contexts of infection. For example, while fibroblasts express high levels of UL138, they also express high levels of UL135, which overcomes the suppressive effects of UL138 [63]. In primary CD34+ HPCs latently infected with viruses encoding epitope-tagged versions of UL133-UL138, the expression of UL138 was detected, but not that of UL135 or UL136 [53]. This may be due, in part, to the distinct transcriptional environment of CD34+ HPCs relative to fibroblasts. For example, the early growth response-1 transcription factor (EGR-1) is critical for the maintenance of stemness and downregulation of EGR1 results in differentiation of stem cells and their migration out of the bone marrow [64]. EGR-1 is highly expressed during infection in CD34+ HPCs, where is serves to drive UL138 gene expression for latency [65]. These observations and others suggest that distinct environments of the host cell play a key role in controlling the balance of HCMV gene expression and the outcome of virus-host interaction in the establishment of latent or replicative infections.

## 4. The HCMV *UL133-138* Polycistronic Locus

*UL133-138* polycistronic locus within the UL*b*’ region encodes viral proteins UL133, UL135, UL136 and UL138 (Figure 1). The transcript encoding all 4 genes is 3.6 kb long, but shorter 3′ co-terminal transcripts are expressed with varying kinetics and under different conditions, such as serum starvation [66,67]. A 2.7 kb transcript spans from *UL135* through *UL138*, and the 1.4kb transcript initiates within the *UL136* gene and encodes truncated forms of UL136 and full-length UL138 [67]. Within 6 h of viral entry into a fibroblast cell, *UL133-UL138* transcripts can be detected and they accumulate as the infection progresses [67]. This coincides with the synthesis of IE1/2, and occurs independently of viral DNA synthesis [67]. Following the onset of viral DNA synthesis, transcripts initiating just 5′ of or within *UL136* accumulate to greater levels, and it is these transcripts that give rise to UL136 proteins [68]. It is currently unknown how the initiation of transcripts might be differentially regulated in infection and what that might mean for HCMV latency, but it would conceivably change the ratios of expression among UL133-UL138 proteins. All UL133-UL138 proteins are predicted to be intrinsically disordered in structure and do not closely resemble any other proteins within the existing databases. In fact, while *UL133-UL138* genes are conserved from HCMV to chimpanzee CMV [69], clear orthologues are not present in rhesus CMV, although the genome of rhesus CMV contains a UL*b*’-like region. No other CMV genomes of viruses infecting lower order animals contain a UL*b*’ region or *UL133-UL138* orthologues [53]. The following sections review the *UL133-UL138* genes and their roles in modulating HCMV latency and/or reactivation.

### 4.1. Pro-Latency Genes of the UL133-138 Locus

#### 4.1.1. UL133

The *UL133* gene marks the 5′ end of the *UL133-138* polycistronic locus (Figure 1), and UL133 is the least characterized protein encoded by the locus. The UL133 protein is 259 amino acids (aa) in length and has a molecular mass estimated at 42 kDa [53]. UL133 has two putative transmembrane domains (TMD), which potentially allow the protein to interact with microsomal membranes by spanning the membrane twice. Upon infection of fibroblasts, UL133 expression is detectable as early as 12 h, and the protein is diffuse at 24 h post infection (hpi), but becomes mostly localized to the Golgi by 48 hpi [53].

UL133 is expressed in CD34+ HPCs at 2 and 5 days post infection (dpi) [53]. Disruption of the expression of UL133 results in a more replicative virus in CD34+ HPCs indicating that UL133 functions in the establishment of HCMV latency. Further corroborating a role in suppressing virus replication for latency, UL133-mutant viruses synthesize viral genomes to higher levels that WT infection [56]. The function of UL133 during HCMV infection may be linked to its interactions with other viral proteins, particularly UL138 and UL136, with which it forms a complex [61,70]. The protein complex may cooperatively favor a latent state of infection since the replication advantage of UL133- and UL138-mutant viruses in CD34+ HPCs are not additive when combined [53,70]. Additional research is required to precisely determine the specific function(s) of UL133.

#### 4.1.2. UL138

UL138 is expressed in all HCMV models of infection, including fibroblasts (highly productive infection), endothelial cells (smoldering low-level productive infection), CD34^+^ HPCs and THP-1 cells (latent infection) [43,53]. *UL138* encodes a protein of 210 aa in length and has a molecular mass estimated at 24 kDa. UL138 can also be expressed as a shorter isoform initiating from methionine codon 16 and lacks the TMD [71]. Both the long and short isoforms contribute to the establishment of latency.

The UL138 proteins are translated from all 3 major transcripts (3.6, 2.7 and 1.4 kb) of the locus (Figure 1), although abundant expression is from the shorter transcripts. A transcriptional element, including an AP-1 binding site, was identified upstream of the transcriptional start of the 1.4 kb transcript and the deletion of this putative promoter element enhanced virus replication in fibroblasts [72]. Although the impact of this deletion on expression of other UL133-UL138 proteins was not determined, the phenotype is consistent with disruption of the UL138 gene or its protein coding potential [57,67]. This is important as the short UL136 isoforms (UL136p23/UL136p19) are also suppressive to virus replication and are expressed from the 1.4 kb transcript [66,68]. UL138 is indispensable for the establishment of latency in CD34+ HPCs as its disruption results in increased virus replication, a failure of the virus to enter a latent state [53,56,67,70]. These results suggest that UL138 acts as a “brake” on viral replication, directing it towards latency. This is further supported by observations that UL138 is suppressive to virus replication to the extent that viruses lacking UL135 (a pro-reactivation protein discussed below) cannot reconstitute infection from the transfection of infectious clones of the HCMV genome, a phenotype that can be rescued by the additional disruption of UL138, and are defective for reactivation [61].

The mechanisms by which UL138 functions to repress virus replication are intriguingly complex. The repressive ability comes partly from preventing the demethylation of histones leading to silencing of the viral genome. UL138 was reported to prevent host lysine demethylases LSD1, JMJD3, and the lysine demethylase-associating protein CtBP1 from interacting with and activating the major immediate early promotor (MIEP) [73]. Activation of the MIEP drives expression of critical viral genes necessary for lytic replication [74,75] and during latency, the MIEP is largely silenced [28,76]. The mechanism by which UL138 contributes to silencing of the MIEP is likely to be indirect since UL138 is associated with secretory membranes. Recently, MIE gene expression for reactivation was found to be driven by promoter elements within the first intron of the MIE transcriptional locus (intronic promoters 1 and 2, iP1, iP2) [43]. It is not yet known if the activity of MIE intronic promoters is influenced by UL138. Further, it was shown that UL138 regulates EGFR signaling, promote EGFR/PI3K/AKT signaling, which is suppressive to viral replication. The role of UL138 in promoting EGFR/PI3K/AKT signaling contributes to viral latency since inhibition of these pathways stimulate virus reactivation in wild type infection, particularly in combination with differentiation stimuli, but does not further enhance replication of a UL138-mutant virus in CD34+ HPCs [59].

UL138 also regulates its own expression through EGFR-stimulated EGR-1, which binds to the HCMV genome upstream of *UL138* and stimulates *UL138* gene expression in both fibroblasts and CD34+ HPCs [65]. When EGR-1 binding is disrupted, UL138 expression relative to that of UL135 falls in CD34+ HPCs and the virus fails to establish latency. By this mechanism, HCMV has hardwired itself into host signaling pathways to “sense” and “respond” to changes in the cellular environment. The HCMV micro RNA, miR-US22, downregulates EGR-1 during infection, which results in decreased *UL138* gene expression [22]. Thus, changes in host signaling or the transcriptional environment of the host influences “decisions” to establish latency or reactivate from latency.

Beyond EGFR, UL138 also enhances levels of tumor necrosis factor-alpha (TNF-α) on the surface of infected cells [77,78], but more studies are required to define a role for UL138-mediated modulation of TNF-α in HCMV latency or in modulation of the innate response to infection. The upregulation of TNFα may poise cells to sense stress of inflammation and cue reactivation. UL138 has also been shown to downregulate the multidrug resistance protein 1 (MRP-1) from the surface of infected cells post-transcriptionally [79]. MRP1 is a transporter that has a broad range of substrates, but one of note is glutathione [80]. The targeting of MRP1 by UL138 increases susceptibility of CD34+ and CD14+ cells to vincristine [81], and while this provides a possible method to target infected cells, it is not clear how the regulation of MRP-1 impacts infection. Golgi sorting motifs in UL138 are required for the downregulation of MRP-1 [81], but they are not important to the establishment of latency [81], suggesting that the regulation of MRP-1 *per se* is not important for latency. Collectively these studies suggest a broad role in regulating the trafficking of several host proteins with as yet unclear roles in infection.

Recently, proteomics approaches identified 170 host proteins differentially regulated in fibroblasts infected with a *UL138*-deletion mutant relative to wild type virus. These include proteins involved in the innate immune response, regulating ubiquitination, and vesicular transport [82]. Innate responses were heightened in cells infected with WT virus relative to a *UL138*-mutant virus, suggesting that UL138 functions to induce the interferon-stimulated genes to repress virus replication to establish latency.

*UL138* gene expression is influenced by other viral factors, including the latency unique natural antigen or LUNA encoded for by the latency-associated transcript UL81-81ast. Deletion of LUNA in fibroblast or CD14+ cells caused a 100- or 1000-fold decrease in UL138 transcript expression, respectively [83]. Further, an HCMV-encoded micro RNA, miR-UL36, was also reported to downregulate UL138 expression in reported assays, but has not been demonstrated to regulate UL138 in the context of infection [84]. Therefore, HCMV may employ multiple mechanisms to regulate UL138 expression, with significance for the establishment of or reactivation from latency.

### 4.2. Genes That Are Pro-Reactivation

#### UL135

The *UL135* gene is expressed from the 3.6-kb and 2.7-kb transcripts [66]. The 2.7-kb transcript initiates downstream of the annotated AUG for initiation of UL135 translation [66]. Further, mutational analysis shows that the annotated AUG corresponding to the methionine 1 start codon (M1) of *UL135* is not used during infection in fibroblasts, and translation initiates at an AUG corresponding to the annotated M21 [63]. UL135 is expressed as two isoforms. The full-length 308-aa isoform is the predominantly expressed isoform in the context of infection. This isoform has a predicted TMD near the N-terminus and it has been demonstrated to be a type-I integral membrane protein that associates with microsomal membranes [53] and it is likely modified post-translationally [63]. The second isoform is soluble and initiates at an AUG corresponding to M97 of the full-length protein. During productive infection, the M21 and M97 isoforms of UL135 have distinct kinetics of expression, where the M97 isoform accumulates at a later times post-infection relative to the M21 isoform [61]. The M97 isoform also lacks the TMD that is present in the M21 isoform [58]. How these isoforms each contribute to infection is currently unknown. UL135 also localizes to the Golgi apparatus and actin cytoskeleton during productive infection in fibroblasts [53,67,85].

UL135 is expressed in all HCMV models of infection: fibroblasts, endothelial cells, CD34^+^ HPCs, and THP-1 cells [43,53]. Similar to *UL138*, *UL135* is expressed early in a productive infection and its expression is independent of vDNA synthesis [67,68]. Interestingly, UL135 has an appreciable role in viral replication across the infections of all three cell types, albeit to varying degrees with context-dependent subtlety in infection outcomes.

Given that the *UL133-UL138* locus is largely dispensable for viral replication in fibroblasts, it was surprising that disruption of *UL135* resulted in a marked defect in initiating replication from the transfection of full-length genomes into fibroblasts [61]. Because the entire locus could be disrupted, but not UL135 alone, it was hypothesized that UL135 was required to overcome suppression to virus replication imposed by another gene within the locus. Indeed, the defect in replication of UL135-mutant viruses could be partially overcome by the simultaneous disruption of UL138, demonstrating the antagonistic relationship between *UL135* and *UL138*. Defects in *UL135*-mutant virus replication in fibroblasts are less apparent when initiating infection from virus particles, as opposed to genome transfection, and at higher multiplicity of infection (MOI).

The antagonism between UL13*5* and UL138 is important to the outcome of infection in CD34+ HPCs. Specifically, *UL135*-mutant viruses replicate viral genomes to lower levels and fail to reactivate in latently infected CD34+ HPCs. Similar to fibroblasts, co-disruption of *UL138* partially restores the observed defects associated with *UL135* disruption; genome replication is restored, but reactivation is less efficient than wildtype infection, suggesting that UL135 has independent replication/reactivation-promoting functions beyond antagonizing UL138 [4,61].

Insight into the mechanisms by which UL135 functions have been gleaned from identifying host interacting partners [58,59,85]. UL135 interacts with EGFR in fibroblasts, a host interaction shared with UL138 [59]. UL135 targets EGFR for turnover, and its disruptions results in increased levels of total and cell surface EGFR. UL135 also interacts with the host adapter proteins Abl interacting protein (Abi)-1, Abi-2, CD2AP, and SH3KBP1 (CIN85) [60,85]. CIN85 and CD2AP bind the same interaction motif. Abi-1, Abi-2, CD2AP and CIN85 function in regulating cytoskeleton architecture and host signaling [86,87,88,89,90], suggesting and interaction between UL135 and various protein signaling complexes of the host cell. Consistent with UL135-mediated turnover of EGFR, Abi-1 and CIN85 recruit the Cbl E3 ubiquitin ligase that targets EGFR for degradation [86,91,92,93,94]. *UL135* mutant viruses containing point mutations within motifs required for interaction with Abi-1 and CIN85/CD2AP are defective in UL135-mediated trafficking and turnover of EGFR [58]. Disrupting interactions between UL135 and Abi-1 or CIN85/CD2AP results in increase EGFR on the surface of both fibroblasts and CD34+ HPCs, and this results in a failure to reactivate from latency in CD34+ HPCs. These findings link the interaction between UL135 and Abi-1 and CIN85/CD2AP and the targeting of EGFR for turnover to reactivation from latency. Consistent with this assertion, inhibition of EGFR or its downstream pathways (PI3K/AKT or MEK/ERK) stimulates reactivation in combination with a stimulus for differentiation and rescues replication of UL135-mutant viruses [59,65].

Beyond EGFR, interactions between UL135, Abi-1 and Abi-2 have been shown to regulate the WAVE2 regulatory complex to remodel the cellular actin cytoskeleton (f-actin) and evade recognition by natural killer and T cells [85]. This model implicates a novel role of UL135 in immune evasion to ensure undetected infection, a role that may be particularly important in reactivation. Furthermore, UL135 has an epistatic relationship with the viral protein kinase UL97 in facilitating the progression of the lytic infection cycle [61,95]. Specifically, the loss of UL97, the only virus-coded serine/threonine protein kinase, has little effect on replication of laboratory strains (lacking UL*b*’ region) but results in a significantly more pronounced defect in low-passage strains [96]. The requirement for UL97 is conferred by UL135 [60,95]. Both UL97 and UL135 are required to drive a second wave of IE2 expression from the MIEP to stimulate viral gene expression and genome synthesis. This induction of IE2 governed by UL97 and UL135 is significant for infection with 10- to 1000-fold defects in replication when either is inhibited [95]. Taken together these findings indicate important replication-promoting functions for UL135 that function to control “decisions” to enter or exit replicative states.

### 4.3. UL136: A Gene with One Hand in Latency and the Other in Reactivation

*UL136* is a 753-bp gene sandwiched between *UL135* and *UL138*, and is expressed with later kinetics than UL135 and UL138 [68]. Five protein isoforms of UL136 with molecular masses ranging from 19- to 33-kDa originate from multiple transcription and, possibly translation initiation start sites [68] (Figure 1). Each of the isoforms will be referred to according to their molecular mass (e.g., UL136p33 for the full-length 33 kDa isoform). A putative TMD comprised of 22 amino acids was identified in silico in the 2 larger isoforms, UL136p33 and UL136p26. The TMD-dependent topology of the 3 isoforms is not definitive [53]. The UL136 isoforms exhibit distinct subcellular localization when overexpressed in fibroblasts, where UL136p33 and UL136p25 are predominantly Golgi localized whereas, UL136p26 is partially localized to the Golgi and UL136p23 and UL136p19 are distributed throughout the cytoplasm [68]. The presence of UL136p26 in the Golgi primarily reflects its synthesis, since it is cleared from the biosynthetic pathway when ongoing translation is inhibited. In the context of infection, all isoforms partially localize with the Golgi, but in this context only UL136p26 is predominantly associated with the Golgi, and the Golgi-cytoplasmic distribution of the other UL136 isoforms is altered from the context where they are expressed transiently. The altered distribution in infection may reflect interactions between isoforms or with other viral proteins.

Expression of UL136 isoforms in fibroblasts follows a kinetic pattern (Figure 2B) characterized by accumulation of UL136p33, UL136 p23/p19 beginning at 24 hpi in high MOI infection. The levels of these isoforms peak by 48–72 hpi, and then decline thereafter [68]. By contrast, UL136p26 and UL136p25 accumulate robustly by 12 hpi and are present stably through the course of infection. While UL133, UL135 and UL138 are expressed with early kinetics, transcripts encoding UL136 isoforms do not accumulate robustly until after the onset of viral genome synthesis and inhibition of viral genome synthesis sharply diminishes the accumulation of UL136 proteins, indicating early-late kinetics. This intriguing observation offers the possibility that UL136 is expressed later to regulate the transition between UL135- or UL138-dominant states of infections. The differential expression patterns and subcellular localization of the UL136 isoforms suggests that each isoform plays a distinct role in infection.

While UL136 and its isoforms are dispensable for replication in fibroblasts, they have replication-promoting (UL136p33 and UL136p26) and -suppressing (UL136p23/UL136p19) functions in endothelial cells [51]. The loss of UL136p33 or UL136p26 results in defects in replication and maturation in endothelial cells. The small soluble UL136p23 and p19 isoforms have not been separated genetically, but a virus lacking both isoforms has a replicative advantage in endothelial cells. Disruption of the UL136p25 produces no phenotypes in endothelial cells. Interestingly, the replication defects or enhancement associated UL136-mutant viruses in endothelial cells largely correlate with the failure to reactivate from or establish latency, as described in further detail below.

The distinct roles of the UL136 isoforms associated with infection in hematopoietic cells are summarized in Figure 2. Viruses lacking UL136p33 and p26 fail to reactivate in CD34+ HPCs or in huNSG mice, indicating that they are both required for reactivation in CD34+ HPCs [51]. By contrast, the 2 small soluble isoforms, UL136p23/UL136p19, suppress virus replication in CD34+ HPCs and huNSG mice. Viruses that lacked UL136p23/UL136p19 replicated but failed to establish latency in CD34+ HPCs as shown by a high frequency of infectious centers pre-reactivation compared to wild-type infection [51]. Similarly, in huNSG mice, viral genomes in the spleen and liver before or after G-CSF-stimulated reactivation were greater than the WT-infection prior to reactivation. The role of UL136p25 isoform in latency is intriguingly context dependent. UL136p25 favors latency in CD34+ HPCs; its absence results in augmented reactivation in CD34+ HPCs. However, in huNSG mice, disruption of UL136p25 results in a failure to reactivate, suggesting a requirement for this isoform in reactivation and dissemination to the liver and spleen [51]. In endothelial cells, UL136p25 appears to enhance the replication-promoting properties of UL136p33 and UL136p26 in the absence of UL136p23/UL13619 [51]. These studies collectively demonstrate that the UL136 isoforms have cooperative and opposing functions in infection and it will be important to understand how the isoforms function together or are differentially regulated in different contexts of infection. The mechanisms by which UL136 functions remain to be defined.

## 5. Mechanisms of *UL133-UL138* Function and the Manipulation of Cell Signaling Pathways for Latency and Reactivation

The complexity and interplay between latently infected cell-extrinsic factors (e.g., immune response, allogenic stimulation, immunocompromisation) and cell-intrinsic factors (e.g., signaling, survival, viral gene expression) has made defining mechanisms of latency and reactivation challenging. The identification of viral determinants that play roles in suppressing the replicative cycle for latency or in promoting reactivation from latency has been an important step forward in defining cell-intrinsic mechanisms of latency and reactivation, which are the focus of this review. However, cell-extrinsic mechanisms are also emerging for UL133-UL138 genes that impact the environment for latency or reactivation and immune evasion during reactivation [82,85].

HCMV infection induces dramatic changes in cellular signaling pathways during infection. During early infection, within minutes, pathways activated by the virus include the hydrolysis of phosphatidylinositol-4,5-bisphosphate and the synthesis of arachidonic acid [97,98]. These perturbations are induced by UV-inactivated virus particles, suggesting that the early signaling pathways are activated by structural components of the HCMV particle. For example, glycoprotein B (gB) activates the interferon response, PDGFR and EGFR pathways, NFκB and atypical PI3K/AKT signaling [98,99,100,101,102,103]. However, as infection progresses many of these pathways are sharply attenuated [59,65]. Other pathways are known to be altered by HCMV including, AMPK and WNT signaling [98,104,105,106,107,108]. The alteration of host signaling pathways allows the virus to manipulate cellular processes affecting gene expression and protein translation, differentiation, stress, intrinsic and innate responses, and metabolism for replicative and latent infections. While viral miRNAs and proteins have been identified that regulate these pathways, providing important mechanistic insights, much remains to be understood about the complexity and nuance of signaling in different contexts of HCMV infection.

Stress or differentiation pathways are typically associated with reactivation from latency [24,29,36,50,109]. But how does the virus “sense” and “respond” to changes in the infected cell to navigate the “decisions” to sustain the latent program or to reactivate? It has long been appreciated that the HCMV genome is replete with host transcription factor binding sites, indicating a co-evolution towards responsiveness to host cues. Interaction between HCMV and EGFR at the cell surface (Figure 3) is important in the stimulation of signaling pathways to traffic its DNA to the nuclei and to induce signaling important for the establishment of latency in CD34+ cells [110]. HCMV infection increases EGFR protein turnover [59], in addition to a transcriptional downregulation of EGFR [111,112], which together results in a 70% reduction in total levels of EGFR. This finding indicates that the virus attenuates EGFR signaling for optimal replication. UL135 and UL138 further regulate EGFR as discussed above. When UL135 is disrupted, EGFR levels increase by ~30% on the cell surface, whereas disruption of UL138 decreases surface EGFR by ~20% [59]. Outside of infection, overexpression of UL135 resulted in a decrease in surface EGFR levels, but overexpression of UL138 alone had no effect, suggesting that additional viral factors work in concert with UL138 to increase EGFR on the cell surface [59].

EGFR signaling post-entry and delivery of the viral genome to the nucleus, constrains the viral replication program in both fibroblasts and CD34+ HPCs. Inhibition of EGFR or its downstream pathways, PI3K/AKT, MEK/ERK and JAK/STAT, diminish viral latency and enhance replication, particularly in combination with stimuli to differentiate [59,65]. Phosphorylation of AKT in this pathway leads to changes in the host transcriptional milieu or activation of the kinase, mTOR, which affect cell proliferation, differentiation, inflammation and survival. PI3K signaling is important for the survival of HCMV-infected monocytes [113,114]. The role of sustained PI3K/AKT signaling for latency is a phenomenon shared with other herpesviruses, such as Epstein Barr virus, in which latent membrane proteins (LMP) 1 and 2a activate PI3K signaling [115,116,117]. Further, herpes simplex virus 1 (HSV-1) VP11/12 indirectly activates PI3K signaling by interacting with Src-kinase family members Grb2, Shc, and p85, and Kaposi sarcoma-associated herpesvirus (KSHV), which uses both its K1 protein and G protein coupled receptor to activate PI3K/AKT signaling [118,119]. In HSV-1 latency in neurons, PI3K/AKT signaling triggered through the NGF-binding to TrkA receptor tyrosine kinase activates PDK1 and this specific signaling pathway is required to sustain latency [120]. The precise requirement for PI3K/AKT signaling in CD34+ HPCs for latency is not yet known.

Recently, Buehler et al. found that UL138-mediated stimulation of EGFR induces EGR1 through MEK/ERK signaling in CD34+ HPCs [62]. EGR1 is highly expressed in CD34+ HPCs to maintain their progenitor phenotype [64], and it was shown that the stimulation of EGR-1 is one way that MEK/ERK signaling contributes to latency [62]. In turn, EGR1 feeds back to stimulate UL138 expression for latency and miR-US22 targets EGR-1 for downregulation, reducing UL138 expression for reactivation [22]. Consistent with these findings, inhibition of EGFR decreases *UL138* expression [110]. Collectively, these findings indicate a regulatory circuit where the viral program of gene expression is at least partially dictated by the signaling and transcriptional environment of the infected cell (Figure 3). Understanding the signalosome important for the establishment of latency in CD34+ HPCs is a key focus of future work.

The MEK/ERK pathway, as activated via RAS and RAF to increase cell motility/migration [121], and is also important for the survival of infected cells. MEK/ERK signaling prevents cell death in HCMV-infected CD34+ cells through the induction of MCL-1 in latently infected CD34+ cells [122,123]. MCL-1 is induced independently of viral gene expression by gB and gH/gL during virus binding and entry. In addition to stimulating MCL-1 to prevent cell death, ERK signaling through Elk-1 decreases levels of the proapoptotic proteins BIM and PUMA. Similar to the initial stimulation of EGFR by virus entry into CD34+ HPCs, these findings demonstrate that very early events are important to establish a cellular environment for HCMV latency.

In addition to enhancing EGFR on the cell surface, UL138 also enhances the expression of tumor necrosis factor alpha receptor (TNFR) at the cell surface (Figure 3), thus sensitizing infected cells to TNF-α signaling [77,78]. As TNF-α stimulates reactivation from latency, inflammation and cell death, this is a rather curious manipulation of host cell signaling by a viral protein well established to drive latency [47,124]. The TNF-α effect of stimulating reactivation was initially observed in latently infected Kasumi cells, a clonal cell line derived from blasts of a patient with myeloperoxidase-negative acute leukemia [125]. Downstream TNF-α signaling stimulates NFκB, which can stimulate MIEP enhancer region to promote viral gene expression [126]. TNF-α has been shown to stimulate IE gene re-expression for reactivation independently of differentiation [124]. It is possible that UL138-driven elevation of TNFR surface levels function to poise latently infected cells to respond to cues for reactivation. It remains to be determined how elevated TNFR might affect the inflammatory state of the infected cell. This presents an interesting model whereby UL138 may establish a multi-pronged system to intercept or filter cellular cues to sustain the latent infection or reactivate from latency.

UL7 is an additional HCMV gene important for viral reactivation that acts as a ligand for the cellular Flt3 receptor [40]. UL7 stimulates PI3K/AKT and MAPK/ERK signaling and induces differentiation of monocytes and CD34+ HPCs. This finding is consistent with the requirement for MEK/ERK signaling for IL-6-driven stimulation of reactivation in DCs and other myeloid or progenitor hematopoietic cell types or lines [122,123,127], but runs contrary to findings that attenuation of PI3K/AKT and MEK/ERK signal reactivation in CD34+ HPCs [59,65]. It will be important going forward to determine if there is a context dependence or timing roles in downregulating or upregulating PI3K/AKT and MEK/ERK signaling for HCMV latency and reactivation. It is possible that early events in reactivation (e.g., UL135) attenuate these pathways only for later events (e.g., UL7) to activate them for differentiation.

US28, one of four HCMV GPCRs, has both ligand-dependent and constitutive signaling activities and has been shown to have complex roles in latency as well as reactivation. Ligand-independent, constitutive US28 signaling activity attenuates MAPK and NFκB signaling, reducing c-fos/AP-1 activation of immediate early viral gene expression but activate immediate early gene expression in response to cellular differentiation [20,21,128]. Reporter assays show that US28 constitutive signaling activity suppressed immediate early gene expression in undifferentiated THP-1 cells, but activates immediate early gene expression in differentiated THP-1 cells [128]. However, it has also been shown in primary CD34+ cells and humanized mouse models that US28 is required for reactivation, but that ligand-dependent signaling is suppressive and required for latency [129]. While more work is required to understand the nuance and context-dependence of US28-mediated signaling to latency and reactivation, it will also be important to understand how US28 might be functioning to synergize with or antagonize other viral determinants of latency and reactivation.

HCMV miRNAs are a powerful mechanism by which to fine-tune expression of viral or cellular gene expression to regulate latency since they are non-immunogenic and play key roles in latency and reactivation. Many HCMV miRNAs target key regulators of host signaling pathways. In addition to miR-US22 targeting EGR-1, miR-UL22A is also required for reactivation and targets SMAD3 to negatively regulate myelosuppressive TGF-β signaling in infected cells [39]. Moreover, miR-US5-1 contributes to more global myelosuppression by targeting the NAB-1 transcriptional suppressor to stimulate TGF-β expression and secretion. This indicates an elegant mechanism by which HCMV-induces myelosuppression, while protecting the ability of infected cells to differentiate.

Beyond host interactions and the regulation of signaling, the relative expression levels of the *UL133-UL138* gene products, particularly those that oppose one another, is another important point of control in regulating latent and replicative states. While the transcripts encoding *UL133-UL138* are 3′ co-terminal, they have distinct 5′ ends (Figure 1). The promoter or regulatory element(s) driving these transcripts have not been thoroughly mapped, but the possibility exists that transcripts encoded by this region may be expressed differentially depending on the context of infection to allow, for example, the accumulation of UL138 over UL135 or the greater accumulation of some UL136 isoforms over others. Some evidence for this regulation exists. In THP-1 cells, the UL138 protein can be detected throughout the time course of infection, but UL135 is silenced with the establishment of latency and is only detected upon reactivation [43]. Further, when EGR-1 binding upstream of *UL138* is disrupted, relative levels of transcripts shift to favor the larger transcripts including UL135 over smaller transcripts encoding replication-suppressive UL136p23/19 and UL138 [65]. Therefore, in contexts such as CD34+ HPCs, where EGR-1 is highly expressed, *UL138* might be expressed to a greater extent relative to *UL135* to favor the establishment of latency. Much remains to be understood about how expression from the *UL133-UL138 locus is* regulated to control the transitions into and out of latency.

## 6. Questions for Future Directions

The polycistronic *133–138* gene locus is specific to CMV strains infecting higher-order primates, reflecting its co-evolution with long-lived primates and the need to regulate specific host biology. Notable strides have been realized in understanding the function of *UL135* and *UL138*, but many questions remain in fully understanding of each of the *UL133–UL138* gene locus products, how their functions are coordinated and regulated, and their mechanisms of action in the infected cell. Going forward, it will be important to understand the nuances in the regulation and timing of PI3K/AKT and MEK/ERK signaling pathways for latency and regulation by the multitude of viral factors targeting these pathways. While it is clear that the regulation of host signaling is critical for establishing or exiting latency, much remains to be understood about how virus manipulation of these pathways impacts the biology of the infected cell with regards to cell survival, differentiation, viral or cellular gene transactivation, protein translation, and the inflammatory response.

## Figures and Tables

**Figure 1 viruses-12-00714-f001:**
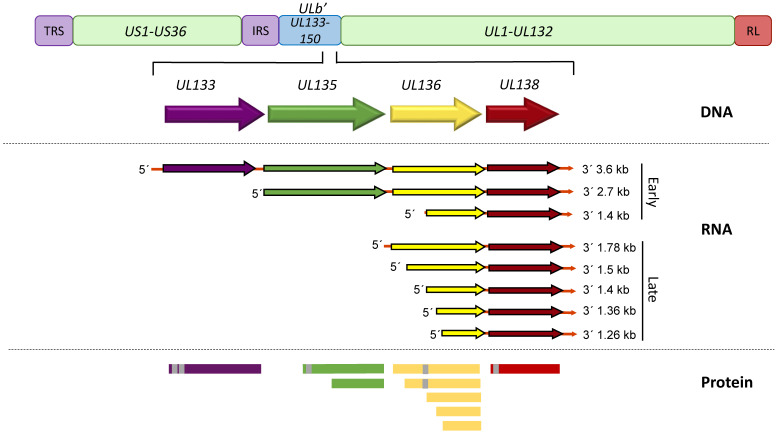
A schematic depiction of the HCMV *UL133-UL138* gene locus. The *UL133-UL138* gene locus, located within the UL*b*’ region, is present in clinical isolates and low passage viruses but lost in serially passaged laboratory strains. The 3 major transcripts that accumulate at early times of infection or additional transcripts encoding UL136 and UL138 that accumulate at late times of infection in fibroblasts are depicted. Alternative transcription/translation initiation sites lead to at least 2 UL135, 5 UL136 and 2 UL138 isoforms. Area shaded grey indicated putative or confirmed transmembrane domains.

**Figure 2 viruses-12-00714-f002:**
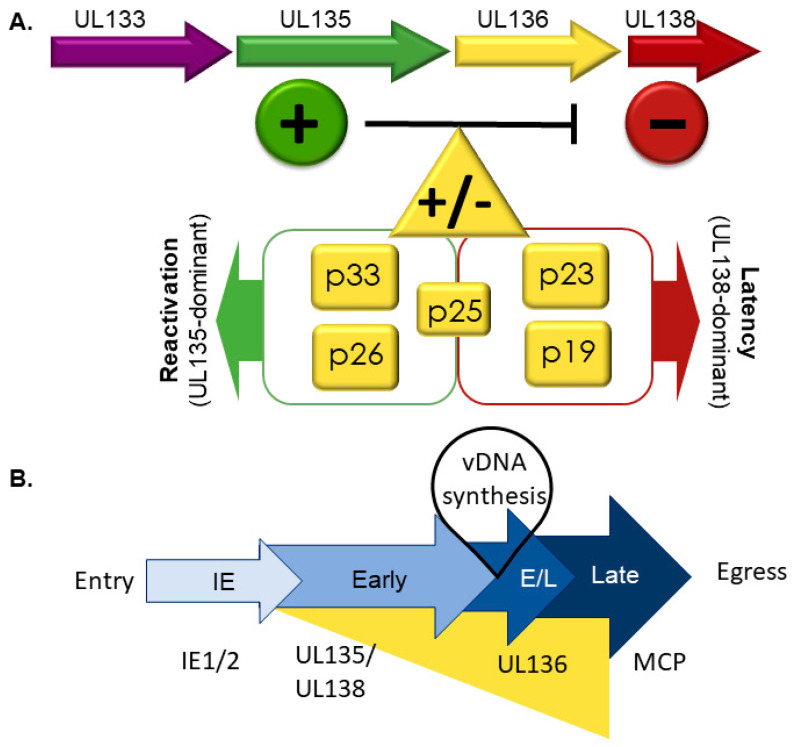
The *UL133-138* gene locus and its effect on latency and reactivation. (**A**) UL138 (red) suppresses virus replication for latency and UL135 (green) is required to overcome these suppressive effects for replication and reactivation. UL136 is expressed as at least five protein isoforms. Like UL133 and UL138, UL136p23/p19 are pro-latency proteins, whereas like UL135, UL136p33 and UL136p26 favor virus replication and are required for reactivation. UL136p25 has context-dependent functions, which need to be further defined. (**B**) Temporal regulation of the *UL133-UL138* genes. Important in this cascade is the differential kinetics of gene expression: early expression of UL135 and UL138 and the early-late expression of UL136 isoforms. Maximal UL136 expression depends on viral genome synthesis, indicating that induction of its expression may function to commit the virus to replicative state or that it mediates the transition between UL138-dominant latent and UL135-dominant replicative states. IE is an acronym for immediate early and E/L denotes early/late.

**Figure 3 viruses-12-00714-f003:**
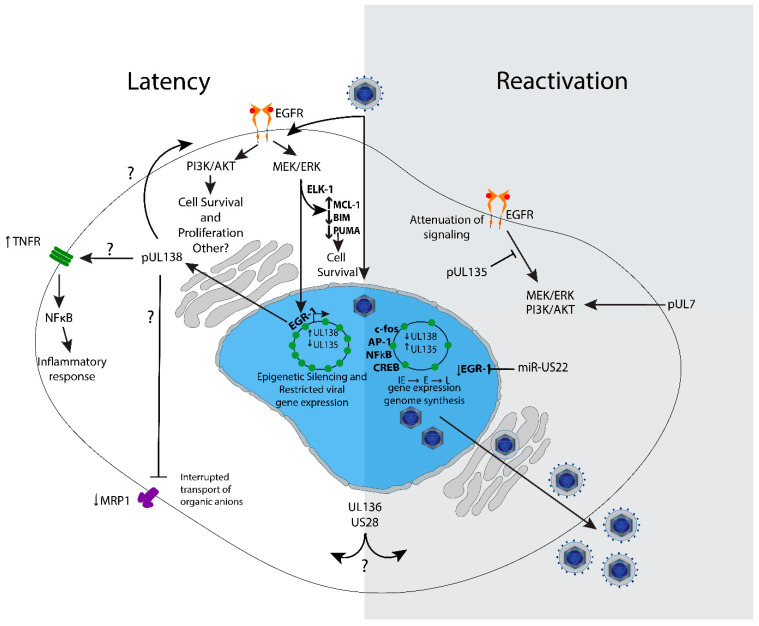
Model of UL133–UL138 virus-host interactions and signaling impacting latency and reactivation. Virus binding and entry stimulates EGFR, PI3K/AKT, and MEK/ERK signaling to set a cellular environment conducive for latency. MEK/ERK stimulates Elk-1-mediated expression of MCL-1 for survival and also downregulates pro-apoptotic Bim and Puma. In latency, the viral genome is repressed, and gene expression is restricted to very low levels. EGFR and downstream PI3K/AKT and MEK/ERK pathways are important for the establishment of latency. MEK/ERK signaling stimulates EGR-1 expression in CD34+ HPCs, which stimulates UL138 gene expression to further enforce the latent infection. UL138 regulates a number of cell surface receptors, including TNFR1, MRP1 and EGFR; the significance of these receptors and their regulation by UL138 is not completely defined, but inhibition of EGFR and its downstream pathways stimulate reactivation and replication. UL135 is expressed upon reactivation. Its interaction with the adapter proteins ABI-1 and CIN85 modulate EGFR lysosomal turnover and cytoskeleton remodeling. The question marks denote a protein function that is currently full defined in the context of latency and needs further study in that context. US28, UL7 and miR-US22 have been show to impact pathways and signaling affected by UL133-UL138. Understanding how these viral factors synergize with or anatomize UL133-UL138 modulation is an important area for future directions.

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
