# Peer review of "The Role of the Human Cytomegalovirus UL133-UL138 Gene Locus in Latency and Reactivation"

_viruses, 2020, doi:10.3390/v12070714_

Round 1
Reviewer 1 Report
Overall this is an excellent review on the functions of the ULb' region of HCMV in lytic infection, latency and reactivation. If anything the article would really benefit from some editing as currently it is somewhat long and repetitive. For example, the antagonistic functions of UL135 and UL138 are mentioned many times throughout the review. How the article is condensed is at the discretion of the authors. My recommendation would be to shorten the sections on the specifics of the proteins themselves and integrate into the section on manipulation of host signaling pathways. The sections on the none ULb' proteins (US28, UL7) could also be shortened to keep the review focused on products of the ULb' genes.
Some minor edits:
Ln 34 of the abstract would be better to say "we and others"
Ln 241 " differences in the cell biology"
Ln 328 - if this refers specifically to the protein it should read translated or synthesized but not expressed.
Ln 402 "The second isoform is soluble initiates at an AUG corresponding the M97." could do with rewriting.
Ln 506-7 "The small soluble UL136p23 and UL136p19 isoforms have not been separated genetically,"
Ln 512 - "Viruses lacking UL133p33 and UL136p26" - should this be UL136p33?
Ln 515 "Viruses that lacked UL136p23/UL136p19 replicated (?but?) failed to establish latency in CD34+ HPCs
Ln's 531-557 - I'm not sure that this paragraph is required and could be removed from the review.
Author Response
Thank you for the constructive comments. We have taken a variety of approaches to shorten the review. All minor comments have been addressed. Thank you!
Reviewer 2 Report
This review article describes the current state of knowledge on the HCMV genes UL133-138 and their importance for latency and reactivation. The review article is from a laboratory that has done groundbreaking work in this field and is very good and comprehensive overall. Nevertheless, there are a few issues that the authors need to address.
Major issues.
- The authors should avoid the use of the first person (we, our laboratory, our research, etc.). A review article should be written from the perspective of a neutral reviewer. Thus, a strong focus on the authors’ own work is inappropriate and should be avoided throughout the manuscript (see also https://researchrundowns.com/writing/writing-a-literature-review ).
- Chapter 2 “Experimental models for HCMV latency and reactivation” is a substantial part of the manuscript and should be mention in the abstract (e.g., In this review, we will describe models of HCMV latency and survey the current understanding … [line 37]). BTW, this is one of the few cases where the use of “we” is appropriate.
- Chapter 3 describes how the Goodrum laboratory got interested in the UL133-138 locus and its role in latency and reactivation. This information may be interesting for family and friends and appropriate for the laboratory webpage, but is not appropriate for a review article. The chapter should be shortened significantly as it contains a lot of redundant information described in detail in the following chapters. The use of “we” and “our” should be avoided.
Minor points.
Line 278. transcripts (not transcript)
Line 387. an HCMV-encoded
Line 402. corresponding to
Line 413. dispensable for
Line 421. Describe the meaning of MOI or use a more general term.
Figure 2b. Describe the meaning of IE and E/L in the legend.
Line 506. have not been separated genetically
Line 541. The advantage of this approach is will identify – correct / rephrase.
Line 601. its K1 and G coupled receptor – rephrase (G protein-coupled?)
Figure 3. increase font size within the figure. Text is barely readable.
Several references contain garbled text, e.g. #33, 45, 51, 76, 78, 81.
Author Response
We have condensed the manuscript and removed all uses of "we" or "our". The review of experimental models is now mentioned in the abstract. All minor revisions have been made.
Reviewer 3 Report
This review on the role of the UL133-UL138 locus in HCMV latency and reactivation is very informative and clear. It is very well written. I only have minor comments:
Line 167: I think it is better to write the word "cells" after Ntera2
Line 188: "... UL151-UL135 was severely..." should be "...were severely..."
Line 243: "mush" should be replaced by "much"
In Figure 2: It is not intuitive that UL133 is a pro-latency protein
Line 515: "Viruses that lacked... replicated failed to establish latency..." I think there is a grammatical error
Author Response
Thank you, all minor comments have been addressed.